# SPP1 Derived from Macrophages Is Associated with a Worse Clinical Course and Chemo-Resistance in Lung Adenocarcinoma

**DOI:** 10.3390/cancers14184374

**Published:** 2022-09-08

**Authors:** Eri Matsubara, Yoshihiro Komohara, Shigeyuki Esumi, Yusuke Shinchi, Shiho Ishizuka, Remi Mito, Cheng Pan, Hiromu Yano, Daiki Kobayashi, Yukio Fujiwara, Koei Ikeda, Takuro Sakagami, Makoto Suzuki

**Affiliations:** 1Department of Cell Pathology, Graduate School of Medical Sciences, Kumamoto University, Kumamoto 860-8556, Japan; 2Department of Thoracic Surgery, Graduate School of Medical Sciences, Kumamoto University, Kumamoto 860-8556, Japan; 3Center for Metabolic Regulation of Healthy Aging, Kumamoto University, Kumamoto 860-8556, Japan; 4Department of Anatomy and Neurobiology, Graduate School of Medical Sciences, Kumamoto University, Kumamoto 860-8556, Japan; 5Department of Respiratory Medicine, Graduate School of Medical Sciences, Kumamoto University, Kumamoto 860-8556, Japan; 6Department of Omics and Systems Biology, Graduate School of Medical and Dental Sciences, Niigata University, Niigata 951-8510, Japan

**Keywords:** adenocarcinoma, GM-CSF, lung, macrophage, SPP1

## Abstract

**Simple Summary:**

Osteopontin, also called secreted phosphoprotein 1 (SPP1), is expressed by cancer cells and is known as a poor prognostic factor. Although the production of SPP1 by tumor-associated macrophages (TAMs) has been attracting much attention recently, there have been no studies distinguishing the SPP1 expression of cancer cells and TAMs. In the present study, we demonstrated the following points. (1) Increased SPP1 expression on TAMs is associated with a worse clinical course in EGFR-wild-type adenocarcinoma. (2) SPP1 expression on macrophages is dependent on GM-CSF-mediated macrophage differentiation. (3) Macrophage-derived SPP1 potentially contributed to chemoresistance in lung cancer.

**Abstract:**

Osteopontin, also called secreted phosphoprotein 1 (SPP1), is a multifunctional secreted phosphorylated glycoprotein. SPP1 is also expressed in tumor cells, and many studies demonstrated that a high level of circulating SPP1 is correlated with a poor prognosis in various cancers. SPP1 is expressed not only by tumor cells but also by stromal cells, such as macrophages. However, there have been no studies distinguishing the SPP1 expression of cancer cells and tumor-associated macrophages (TAMs). Thus, in this study, we tried to accurately evaluate the SPP1 expression status on cancer cells and TAMs separately in patients with non-small cell lung cancer by using double immunohistochemistry. We demonstrated that high SPP1 expression on TAMs predicted a poor prognosis in lung adenocarcinoma patients. Additionally, we investigated the expression mechanisms related to SPP1 using human-monocyte-derived macrophages and revealed that the SPP1 expression level increased in macrophage differentiation mediated by granulocyte-macrophage colony-stimulating factor. Furthermore, SPP1 contributed to anti-cancer drug resistance in lung cancer cell lines. In conclusion, SPP1 production on TAMs predicted a poor prognosis in lung adenocarcinoma patients, and TAM-derived SPP1′s involvement in the chemo-resistance of cancer cells was suggested.

## 1. Introduction

Lung cancer is the most common cause of cancer death in the Japanese population, and its incidence increases sharply after the age of 50 years [1,2]. Although recent advancements and the widespread use of computed tomography scanning contributed to the early diagnosis of lung cancer, about two-thirds of lung cancer patients are still diagnosed at advanced stages [3,4]. Non-small cell lung cancer (NSCLC) accounts for about 85% of all lung cancers [5]. The treatment options for NSCLC depend mainly on the stage of the cancer. Surgery is an option for early-stage NSCLC, and it provides the best chance to cure the disease. However, postoperative recurrence often occurs even in cases that were able to receive radical resection [1,2,3].

Osteopontin, also called secreted phosphoprotein 1 (SPP1), is a multifunctional secreted phosphorylated glycoprotein that has an arginine-glycine-aspartate-containing (RGD-containing) domain, which was first identified in bone tissue as a major sialoprotein that modulates bone formation and remodeling. The RGD site of SPP1 can bind to multiple integrins, such as α_v_β_3_, α_v_β_5_, α_v_β_1_, and α_5_β_1_, and to certain variant forms of CD44 [6,7,8,9].

Previous studies demonstrated that the level of circulating SPP1 and/or increased SPP1 expression on tumor cells were correlated with a poor prognosis in various cancers, including NSCLC [7,10,11,12,13]. In general, analyses for detecting SPP1 can be categorized into two subgroups: enzyme-linked immunosorbent assay (ELISA) of blood samples and immunohistochemistry (IHC) of cancer tissue samples. In almost all reports using IHC, lung adenocarcinoma and SCC were evaluated interchangeably [14,15,16].

SPP1 is expressed not only by tumor cells, but also by stromal cells, such as macrophages [17,18]. Macrophages that infiltrate in cancer microenvironment are referred to as tumor-associated macrophages (TAMs). TAMs have protumor functions related to neovascularization, invasion, and immunosuppression, and an increased density of TAMs has been shown to be associated with a poor clinical course in many cancers, including lung cancer [19,20,21]. Although the production of SPP1 by TAMs has been attracting much attention recently, there have been no studies distinguishing the SPP1 expression of cancer cells and TAMs. Thus, in this study, we tried to accurately evaluate the SPP1 expression status on cancer cells and TAMs separately in patients with NSCLC by double-IHC, and demonstrated that only SPP1 expression on TAMs predicted a poor prognosis in lung adenocarcinoma patients.

## 2. Materials and Methods

### 2.1. Samples

Paraffin-embedded tissue samples were from 228 patients diagnosed with lung adenocarcinoma and 103 patients diagnosed with lung squamous cell carcinoma between 2008 and 2013 at Kumamoto University Hospital, Japan. All patients underwent surgical resection and achieved complete resection (R0). The staging was based on the most recent IASLC TNM classification system [22]. The clinicopathological data of the patients are summarized in Table 1. Two pathologists reviewed all tissue specimens, and the most representative area of a 5 mm-diameter core containing viable cancer cells was selected for tissue microarrays.

### 2.2. In Situ Hybridization (ISH)

In situ hybridization was performed using the RNAscope^Ⓡ^ 2.5 HD Duplex Detection Kit (#322500-USM; Advanced Cell Diagnostics, Newark, CA, USA) according to the manufacturer’s instructions. The RNAscope2.5 Diplex Positive Control (#321641; Advanced Cell Diagnostics) was used as the positive control probe, and the RNAscope 2-Plex Negative Control Probe (#320751; Advanced Cell Diagnostics) was used as the negative control probe.

### 2.3. IHC

Paraffin sections were subjected to single and double IHC using a routine protocol that has previously been published [23,24,25]. In brief, anti-SPP1 antibody (AF1433; R&D Systems, Minneapolis, MN, USA), anti-Iba-1 antibody (Wako, Tokyo, Japan), anti-CD204 (clone SRA-E5; CosmoBio, Tokyo, Japan), and anti-PU.1 (clone EPR3158Y; Abcam, Cambridge, UK) were used as the primary antibodies. Anti-PU.1 is used as a pan-macrophage marker and it is positive in the nucleus [26]. All immunostained sections were evaluated by two investigators (Y.K. and E.M.). The SPP1 expression level on cancer cells and macrophages was scored according to the proportion of stained cells as follows: less than 1% staining, score 0; 1% to 49% staining, score 1; more than 50% staining, score 2.

### 2.4. Cell Culture of Macrophages and Cancer Cell Lines

The human lung adenocarcinoma cell lines NCI-H358, H23, and H1975 were obtained from Tomoya Yamaguchi (Kumamoto University, Kumamoto, Japan). The other human lung adenocarcinoma cell lines (PC-9, LC-KJ, and A549) and human lung squamous cell lines (LC-1, LK-2, VMRC, and KNS-62) were obtained from the Japanese Collection of Research Bioresources Cell Bank (Osaka, Japan). THP-1 and U937 cell lines were obtained from American Type Culture Collection (Manassas, VA, USA). All cancer cell lines were cultured in DMEM or RPMI (Wako) with 10% fetal bovine serum. THP-1 cells were supplemented with phorbol-12-myristate-13-acetate to induce differentiation. Human monocyte-derived macrophages were obtained from healthy donors and cultured as described previously in accordance with protocols approved by the Kumamoto University Hospital Review Board (no. 1169) [23,24,25,27]. These monocytes were plated in UpCELL 6-well plates (5 × 10^5^ cells/well; CellSeed, Tokyo, Japan) and cultured in AIM-V medium (Thermo Fisher, Waltham, MA, USA) supplemented with 2% human serum, granulocyte-macrophage colony-stimulating factor (GM-CSF; 1 ng/mL; Wako), and macrophage colony-stimulating factor (M-CSF; 100 ng/mL; Wako) to induce macrophage differentiation. For the preparation of the conditioned medium (CM) of cell lines or macrophages, cancer cells (80% confluent) or differentiated macrophages cultured new medium for 1 day, and supernatants were collected. SPP1, Paclitaxel (PTX), and pemetrexed (PEM) were obtained from FUJIFILM Wako Pure Chemical Corporation (Osaka, Japan). SPP1 was used at a concentration of 400 ng/mL. For the WST-8 assay, viable cells in each well were quantified by using Cell Counting Kit-8 (DOJINDO, Kumamoto, Japan). PTC-209, the BMI1 inhibitor, was purchased from Selleck Bioteck (Tokyo, Japan), and dissolved according to the manufacturer’s instructions.

### 2.5. 3D Cell Culture

H23 or H1975 cells were treated with PTX or PEM (10 µM) for 3 days. Cells were washed and seeded on 96-well plates (1 × 10^3^ cell/well, Costar^Ⓡ^ 96-well Ultra Low Attachment Multiple Well Plate, flat bottom, Corning, Glendale, AZ, USA) with SphereMax (Nissan Chemical, Omiya, Japan). After incubation for 10 days, the viable cells in each well were quantified using CellTiter-Gro^Ⓡ^ Luminescent Cell Viability (Promega, Fitchburg, WI, USA).

### 2.6. Quantitative Real-Time Reverse Transcription PCR (qRT-PCR) Analysis

Total RNA was isolated from cells using RNAiso Plus (Takara Bio, Shiga, Japan). qRT-PCR was performed with TB Green Premix Ex Taq II (Takara Bio) on an Applied Biosystems 7300 Real-Time PCR System (Applied Biosystems, Tokyo, Japan). A minimum of three individual samples was used, and the expression levels were calculated from at least two technical replicates. The mRNA expression levels were estimated using the 2ΔΔCt method, and the mRNA levels were normalized those of β-actin mRNA. All primers were purchased from TAKARA (Shiga, Japan), and these were pre-designed. The primers are listed in Appendix A.

### 2.7. Western Blot Analysis

The cellular proteins were solubilized in a Tris buffer containing 2% sodium dodecyl sulfate and 10% glycerol. The following antibodies were used for Western blotting: anti-SPP1 (AF1433; R&D Systems), anti-CD204 (clone SRA-E5; CosmoBio), anti-Bcl-2 (D17C4; Cell Signaling Technology, Danvers, MA, USA), anti-Bax (D2E11; Cell Signaling Technology), anti-cleaved caspase-3 (D175; Cell Signaling Technology), anti-cleaved PARP (Asp214; D64E10; Cell Signaling Technology), anti-Bmi1 (EPR3745(2); abcam, Cambridge, UK), and β-actin (sc-47778; Santa Cruz Biotechnology, Dallas, TX, USA). The detected bands were quantitatively measured using Image J [28]. The original WB can be found in the Appendix A.

### 2.8. ELISA for SPP1

ELISA for SPP1 was performed using the Human Osteopontin Assay Kit-IBL (#27158; Immuno-Biological, Gunma, Japan) according to the manufacturer’s instructions.

### 2.9. RNA-Sequencing (RNA-Seq) Transcriptome Analysis

H23 cells were treated with the CM of macrophages and SPP1 for 1 day 24 h; then, total RNA was isolated with RNAiso Plus (Takara Bio). The total RNA was prepared by Trizol extraction (Thermo Fisher), and the quality was confirmed using BioAnalyzer 2100 (RNA integrity number > 9). For the RNA-seq data analysis, the resulting reads were aligned to the human genome (UCSC hg19) using STAR ver.2.6.0a (Dr. Alexander Dobin, Cold Spring Harbor Laboratory, NY, USA) after trimming to remove adapter sequences and low-quality ends using Trim Galore! v0.5.0 (cutadapt v1.16) (Babraham Bioinformatics, Cambridge, UK) [29]. Gene expression levels were measured in terms of transcripts per million as determined by RSEM v1.3.1 (Dr. Bo Li and Dr. Colin Dewey, University of Wisconsin-Madison, Madison, WI, USA) [29].

### 2.10. Statistical Analysis

Statistical analysis was carried out using JMP7 software (SAS Institute, Chicago, IL, USA), Prism software (GraphPad Software, San Diego, CA, USA), and EZR [30]. Pearson’s chi-square test or Fisher’s exact test was used to analyze the correlation between SPP1 expression and clinical pathologic features. Cancer-specific survival (CSS) was defined as the time from surgery to death for cancer. Progression-free survival (PFS) was defined as the time from surgery to occurrence of cancer progression or death. CSS and PFS were analyzed using Kaplan–Meier methods and compared by the Log-rank test and Wilcoxon test. Multivariate analyses were performed by using Cox‘s proportional hazard model. The comparisons between two groups were analyzed with a Student’s *t*-test, or a Mann–Whitney U test. Spearman’s correlation analysis was used for correlation analysis. For all analyses, *p* < 0.05 was considered to indicate statistical significance.

## 3. Results

### 3.1. SPP1 Is Expressed on TAMs Infiltrating the Tumor, but Not on Alveolar Macrophages in Non-Tumor Areas

To evaluate the protein and mRNA expression levels of SPP1, ISH and IHC were performed using serial sections from adenocarcinoma and SCC cases. Dual-color-ISH revealed that SPP1 mRNA was expressed on Iba-1-positive macrophages (Figure 1), and similar results were obtained from IHC of SPP1 and macrophage markers (Iba-1 and CD204; Figure 1). Notably, SPP1 was more highly expressed on TAMs infiltrating the tumor than on macrophages in non-tumor areas around the tumor (Figure 2A). SPP1 was positive for foamy-like macrophages existing in the tumor nests or glandular area, but SPP1 was less expressed on smaller-sized macrophages in the stroma (Figure 2B). Therefore, we focused on the foamy-like macrophages existing in the tumor nests and glandular area for further analysis. However, SPP1 was strongly positive in both the cancer cells and TAMs in some cases (Figure 2C). Therefore, to evaluate the SPP1 expression level on cancer cells and TAMs separately, double-IHC using macrophage markers was performed. To examine the SPP1 expression on cancer cells, a double-IHC of SPP1 and Iba-1 was performed (Iba-1-negative; Figure 2D). To examine the SPP1 expression on TAMs, the double-IHC of SPP1 and PU.1 was performed since the signals of SPP1 and Iba-1 overlapped (PU.1-positive in the nucleus; Figure 2D). The proportions of samples with each score among cancer cells and TAMs in adenocarcinoma and SCC are shown in Figure 2E. In adenocarcinoma, the chi-squared test showed no relationship between the score for SPP1 on TAMs and age, sex, smoking status, pStage, or the presence of an epidermal growth factor receptor (EGFR) mutation (Table 2). In SCC, only younger age groups less than 65 years of age tended to have a higher score for SPP1 on TAMs (Table 3). In contrast, the scores for SPP1 on cancer cells were not correlated with any factor (Appendix A).

### 3.2. Increased SPP1 Expression on TAMs Is Associated with a Worse Clinical Course in EGFR-Wild-Type Adenocarcinoma

In adenocarcinoma, the scores for SPP1 on cancer cells were not correlated with the prognosis, whereas high scores for SPP1 on TAMs were found to predict a poor prognosis (Figure 3A). Cancer-specific survival (CSS) was significantly shorter in the group with high scores for SPP1 on TAMs. As for the progression-free survival (PFS), a similar trend was observed, but not to a statistically significant level (Figure 3A). However, when the analysis was limited to the EGFR-wild-type cases, CSS and PFS were significantly shorter in the group with high scores for SPP1 on TAMs (Figure 3B). In contrast, in SCC, the scores for SPP1 on cancer cells and TAMs were not correlated with the prognosis (Appendix A). The CSS and PFS depending on the disease stage in adenocarcinoma are shown in Appendix A.

The univariate Cox regression analysis revealed that some factors, including male sex, smoking, a high pathological stage, the EGFR-wild-type status, and a high score for SPP1 on TAMs, were associated with a shorter CSS in adenocarcinoma. In the multivariate analysis, a high pathological stage, the EGFR-wild-type status, and a high score for SPP1 on TAMs were independent prognostic factors (Table 4). As for the PFS, male sex, smoking, and a high pathological stage were associated with a shorter PFS in the univariate analysis, and only a high pathological stage was an independent factor in the multivariate analysis (Table 5).

### 3.3. SPP1 Expression on Macrophages Is Dependent on GM-CSF-Mediated Macrophage Differentiation

As shown in Figure 1 and Figure 2, in many cases, the SPP1 expression level was higher on TAMs than on cancer cells. Subsequently, the SPP1 expression level in various cells, including macrophages and cancer cell lines, was examined by qRT-PCR. SPP1 was more highly expressed in macrophages and THP-1 (a monocyte-like cell line) than in the lung cancer cell lines (Figure 4A). As it is well-known that cancer cell-derived factors stimulate macrophage activation, we tested whether the CM of cancer cell lines influenced the SPP1 expression in monocyte-derived macrophages and THP-1 macrophages. However, no changes were seen in SPP1 expression in CM-stimulated macrophages (Figure 4B).

Next, SPP1 expression was evaluated at various stages of macrophage differentiation from monocytes. Although no SPP1 expression was observed in monocytes, SPP1 expression was detected in macrophages cultured for 3 and 5 days (Figure 4C). Furthermore, the SPP1 expression level was higher in macrophages differentiated with GM-CSF than in macrophages differentiated with M-CSF (Figure 4D). The SPP1 concentration in the CM of macrophages was examined by ELISA, and we found that the pattern of SPP1 production in the CM was the same as the pattern of SPP1 mRNA expression (Figure 4E). Immunocytostaining of SPP1 in macrophages cultured with M-CSF or GM-CSF at days 0 and 5 showed that only macrophages cultured with GM-CSF for 5 days were positive for SPP1 (Figure 4F). The SPP1 expression levels in THP-1 cells differentiated for 0, 1, 3, or 6 days were examined by qRT-PCR (Figure 4G) and Western blot analysis (Figure 4H), and the results were similar to macrophages.

We also investigated whether there was a correlation between SPP1 gene expression and various macrophage markers, such as CD204, Iba-1, CD68, CD163, and CD163L1, in lung adenocarcinoma tissues. We used gene expression data from a lung adenocarcinoma cohort in The Cancer Genome Atlas (https://www.proteinatlas.org/ (accessed on 16 June 2021)). The strongest correlation was found between SPP1 gene expression and CD204 gene expression in lung adenocarcinoma tissues (Rho = 0.46, *p* < 0.001; Figure 5). In vitro studies using monocyte-derived macrophages and THP-1 macrophages showed that CD204 expression was dependent on macrophage maturation and GM-CSF-stimulation as well as SPP1 (Figure 4C,D,G,H).

### 3.4. Macrophage-Derived SPP1 Potentially Contributes to Chemoresistance in Lung Cancer

We investigated the effects of the macrophage-derived SPP1 on lung adenocarcinoma cells. Since cancer cell proliferation was not influenced by SPP1 and macrophage-derived factors in a two-dimensional (2D) culture system (unpublished data), we focused on the chemoresistance of cancer cells. For cell culture studies, we used the lung cancer cell lines H23 and H1975, which do not express SPP1 (Figure 4A). A 2D cell culture study using PTX and PEM was performed, as shown in Figure 6A. Both PTX and PEM significantly inhibited cancer cell proliferation; however, the cell viability of H23 cells was improved by the CM of macrophages and SPP1 (Figure 6B,C). The CM of macrophages and SPP1 also improved the cell viability of H1975 cells when the cells were treated with PEM (Figure 6C).

Next, we examined whether the CM of macrophages and SPP1 would influence the re-growth of cancer cells pre-treated with PTX and PEM using a three-dimensional (3D) cell culture system (Figure 6D). The CM of macrophages, but not SPP1, significantly promoted cancer cell growth with or without the anti-cancer drugs (Figure 6E,F and Appendix A). Notably, SPP1 alone as well as the CM of macrophages alone promoted cancer cell re-growth (Figure 6E,F and Appendix A). Next, we examined whether the macrophage-derived SPP1 suppresses apoptosis of cancer cells under the administration of anticancer drugs. First, we investigated how long it would take to proceed to the apoptosis pathway when PTX is added to H23. Figure 6G shows that the apoptosis pathway progressed 24 h after PTX administration. Next, H23 cells were cultured with the CM of macrophages or SPP1, and treated with PTX for 24 h; then, the expression levels of apoptosis-related molecules were examined by Western blot analysis. The CM of macrophages and SPP1 induced Bcl-2 expression and reduced apoptosis-related molecules (Figure 6H). We revealed that the macrophage-derived SPP1 suppresses the apoptosis of cancer cells under the administration of anticancer drugs.

### 3.5. BMI1, Potentially Induced by TAM-Derived SPP1, Was Considered to Be One of the Factors Involved in Chemoresistance

We investigated genes that were found to be upregulated by TAM-derived SPP1 in the RNA-seq analysis. The upregulated genes in the cells treated with the CM of macrophages and SPP1 are listed in Figure 7A and Appendix A. We focused on B-cell-specific Moloney leukemia virus insertion site 1 (BMI1), since the BMI1 gene is known to be a poor prognostic factor in several cancers [31,32,33,34]. We confirmed that BMI1 was upregulated at both genetic and protein levels in the cells treated with the CM of macrophages and SPP1 (Figure 7B,C).

We next tested if the overexpression of BMI1 in H23 cells could induce chemoresistance using PTC-209, the BMI1 inhibitor. H23 cells were treated with the indicated concentration of PTC-209 for 48 h, and we confirmed that the protein expressions of BMI1 were downregulated (Figure 7D). We also investigated that what concentration of PTC-209 causes cytotoxicity by WST-8 assay and revealed that cell damage does not occur at concentrations of PTC-209 of 0.5 µM (Figure 7E). Figure 7F shows that the IC50 of PTX relative to H23 cells decreased under the administration of PTC-209.

## 4. Discussion

SPP1 protein is produced in multiple organs throughout the body in physiological and disease conditions, and osteoblasts, fibroblasts, activated macrophages in healing wounds, and dendritic cells are known to be sources of SPP1 [6,9]. Many reports suggested that the levels of circulating SPP1 and/or an increased level of SPP1 expression in tumor cells are correlated with a poor prognosis in NSCLC [7,14,15,16]. Although the production of SPP1 by TAMs has been attracting much attention recently [17,18], there have been no studies distinguishing the SPP1 expression of cancer cells and TAMs. Thus, in this study, we tried to accurately evaluate SPP1’s expression status on cancer cells and TAMs separately in patients with NSCLC by double-IHC and demonstrated that only SPP1 expression on TAMs predicted a poor prognosis in lung adenocarcinoma patients. Additionally, we investigated expression mechanisms related to SPP1 on macrophages. We revealed that the expression of SPP1 increased as the macrophages differentiated and that GM-CSF played a key role in this process. Finally, we revealed that TAM-derived SPP1 contributed to chemo-resistance.

In this study, the SPP1 level on TAMs was found to be a predictor of a poor prognosis in lung adenocarcinoma. The CSS was significantly shorter in the group with high scores for SPP1 on TAMs. As for the PFS, a similar trend was observed, but not to a statistically significant level. As a reason for that, we presumed that the TAM-SPP1-high group was resistant to treatments after recurrence. However, we did not analyze that in detail, and it is unclear. When the analysis was limited to EGFR-wild-type cases, CSS and PFS were significantly shorter in the group with high scores for SPP1 on TAMs. In contrast, in EGFR-mutation-positive cases, the scores for SPP1 on TAMs were not correlated with the prognosis. This might have been due to the high efficacy of EGFR-tyrosine kinase inhibitor treatments for EGFR-mutation-positive cases.

SPP1 expression is regulated by various transcription factors [6]. Choi SI et al. reported that transmembrane 4 L six-family-member 4-triggered SPP1 expression is involved in the persistent intensification of epithelial-mesenchymal transition or cancer stemness by creating a positive feedback autocrine loop with the JAK2/STAT3 or FAK/STAT3 pathways [35]. SPP1 has multiple receptors, such as α_v_β_3_, α_v_β_5_, α_v_β_1_, and α_5_β_1_, and certain variant forms of CD44, which contributes to the complexity of the SPP1 expression mechanism [36]. However, most reports on SPP1 expression mechanisms have been related to the expression of SPP1 in cancer cells, and there have been few reports on SPP1 expression in macrophages. Shirakawa K et al. reported that the interleukin (IL)-10-STAT3-galectin-3 axis is important for SPP1-producing reparative macrophage polarization after myocardial infarction [37]. Shimodaira T et al. reported that IL-6 from fibroblasts stimulated by IL-1β predominately increases the expression of SPP1 in THP-1 macrophages [38]. In the present study, GM-CSF enhanced SPP1 production in monocyte-derived macrophages. Since it is well-known that GM-CSF is linked to STAT3 activation, STAT3 may be a key molecule that regulates SPP1 production in macrophages.

Wang M et al. reported that radiation resistance in KRAS-mutated lung cancer is conferred by stem-like characteristics mediated by the SPP1-EGFR pathway [39]. Wei J et al. reported that SPP1 is a chemokine that causes the infiltration of macrophages into glioblastoma multiforme and that it plays a role in the immune-suppressive properties of macrophages [8]. In this study, TAM-derived SPP1 contributed to the chemo-resistance of lung cancer cell lines, and it additionally promoted re-growth after chemotherapy. Li Y et al. reported that TAM-derived SPP1 upregulated PD-L1 expression in NSCLC cells, and it causes poor prognosis [40]. We previously reported the PD-L1 expression in cancer cells and TAMs separately [23,24] by using same cohort of lung cancer cases. We tested if SPP1 expression of TAMs would be associated with the PD-L1 expression of cancer cells and TAMs in this lung cohort. However, there was no correlation between the SPP1 expression of TAMs and the PD-L1 expression of TAMs or cancer cells in lung adenocarcinoma (Appendix A).

Our RNA-seq results showed that many molecules, including BMI1, were increased by TAM-derived SPP1. BMI1 is overexpressed in several human cancers, and known to be a poor prognostic factor [31,32,33,34]. The inhibition of BMI1 can elevate the sensitivity of tumor cells to chemoradiotherapy for tongue and breast cancer [31,32]. As for lung cancer, Shen HT et al. reported that BMI1 is involved in PEM resistance and that BMI1 overexpression activates epithelial–mesenchymal transitions and enhances cancer stemness in NSCLC [33]. Sugihara H et al. reported that miR-30e* mediated by TAMs directly regulates BMI1 expression in gastrointestinal cancer [34]. Our results may be explained by the fact that TAM-derived SPP1 induces BMI1 in lung cancer cells. Furthermore, BMI1 was considered to be one of the factors involved in chemoresistance.

A recent study using single-cell RNA-seq in human lung cancer revealed that monocyte/macrophage cells could be divided into four groups (Figure 8) [41]: Group I cells are tissue-resident macrophages that express high levels of cell-cycle-related genes; group III cells are CD14-positive monocytes; group IV cells are CD16-positive monocytes; the remaining cells are broadly classified into group II, which are considered to be derived from circulating monocytes, since these cells express mature macrophage genes and lack transcripts expressed by group I cells. In our dataset, SPP1 was found to be strongly expressed by group II cells (Figure 8). Thus, SPP1 may be a marker that can be used to categorize TAMs as tissue-resident cells or monocyte-derived cells.

## 5. Conclusions

SPP1 on TAMs was associated with a poor clinical course in lung adenocarcinoma patients, and the protumor function of SPP1 is potentially related to the chemoresistance and re-growth of cancer cells. Thus, SPP1 derived from TAMs may be a target for anti-cancer therapy in lung adenocarcinoma patients. In addition, SPP1 is a potential marker for identifying monocyte-derived TAMs in lung cancer.

## Figures and Tables

**Figure 1 cancers-14-04374-f001:**
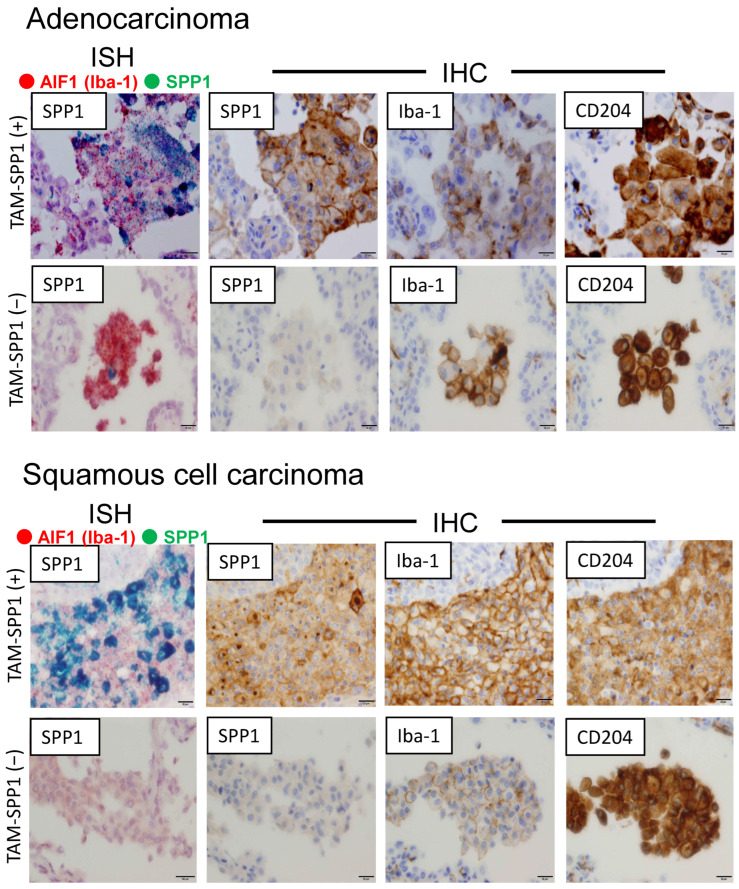
Dual-ISH of SPP1 and Iba-1 (green: SPP1; red: Iba-1) and IHC of SPP1 and other macrophage markers (Iba-1 and CD204) using serial sections of adenocarcinoma and SCC.

**Figure 2 cancers-14-04374-f002:**
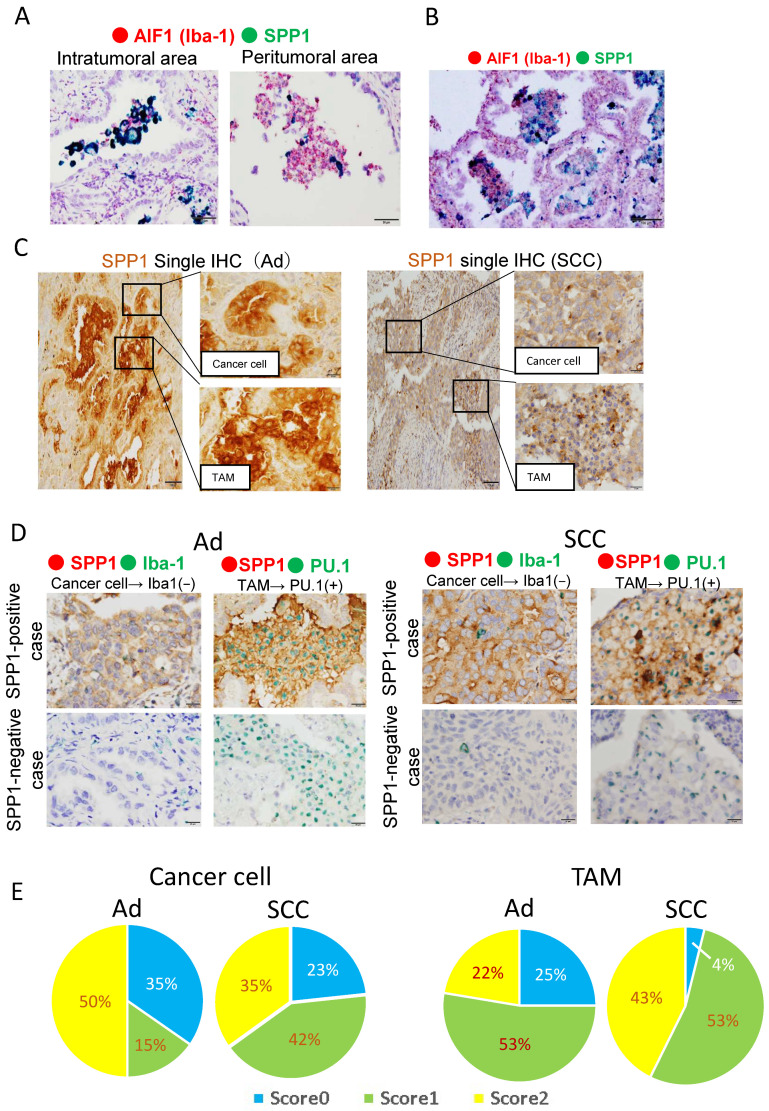
Evaluation of SPP1 expression on cancer cells and macrophages. (**A**,**B**). Representative dual ISH (green: SPP1; red: Iba-1) in adenocarcinoma. (**C**). Representative single-IHC of SPP1 in adenocarcinoma and SCC. (**D**). Double-IHC of SPP1 and Iba-1 to examine the SPP1 expression level in cancer cells (Iba-1-negative). Double-IHC of SPP1 and PU.1 to examine the SPP1 expression level on TAMs (PU.1-positive in the nucleus). (**E**). SPP1’s expression level was scored according to the proportion of stained cells as follows: less than 1% staining, score 0; 1% to 49% staining, score 1; more than 50% staining, score 2. The proportions of samples with each score among cancer cells and TAMs in adenocarcinoma and SCC are shown in the pie charts.

**Figure 3 cancers-14-04374-f003:**
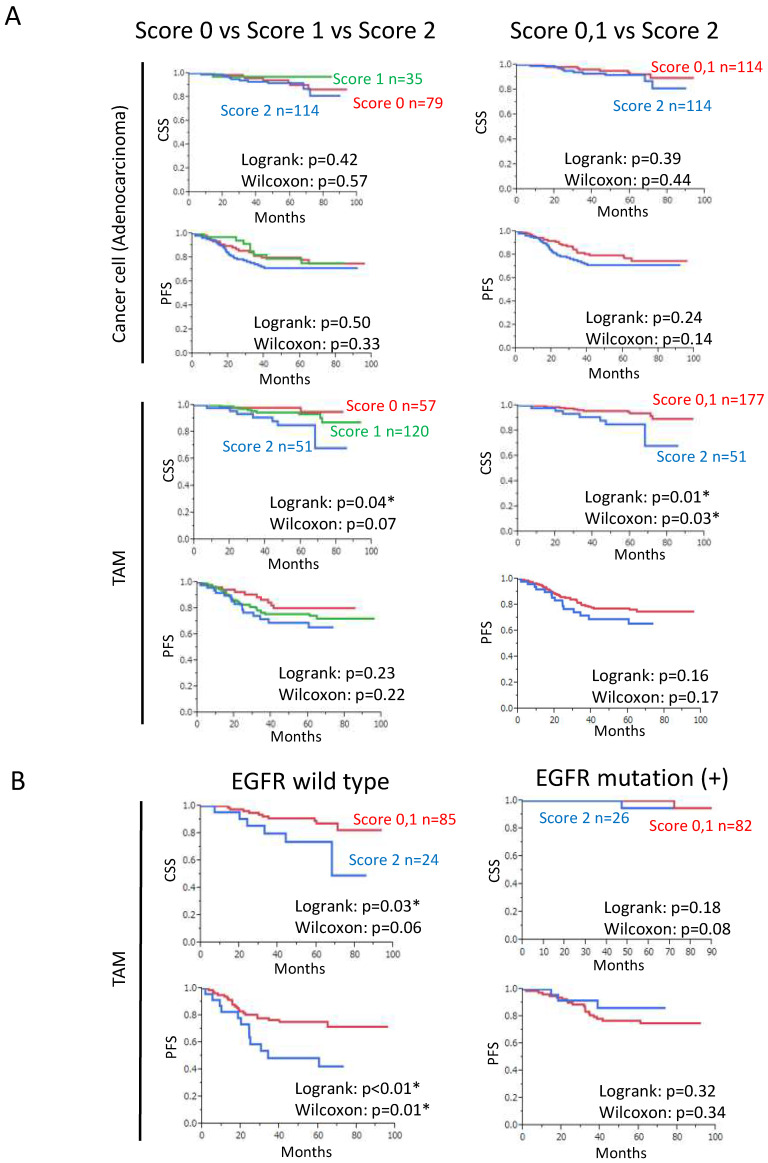
Kaplan–Meier survival analyses according to the SPP1 score in lung adenocarcinoma. (**A**) Kaplan–Meier analyses of the CSS and PFS according to the SPP1 score in cancer cells or TAMs. The graphs on the left show the survival curves for score 0, score 1, and score 2. The graphs on the right show the survival curves for scores 0 and 1 combined, and score 2. (**B**) Kaplan–Meier analysis of the CSS and PFS according to the SPP1 score in TAMs when divided into EGFR-wild-type and EGFR-mutation-positive groups. *: statistically significant, *p* value < 0.05.

**Figure 4 cancers-14-04374-f004:**
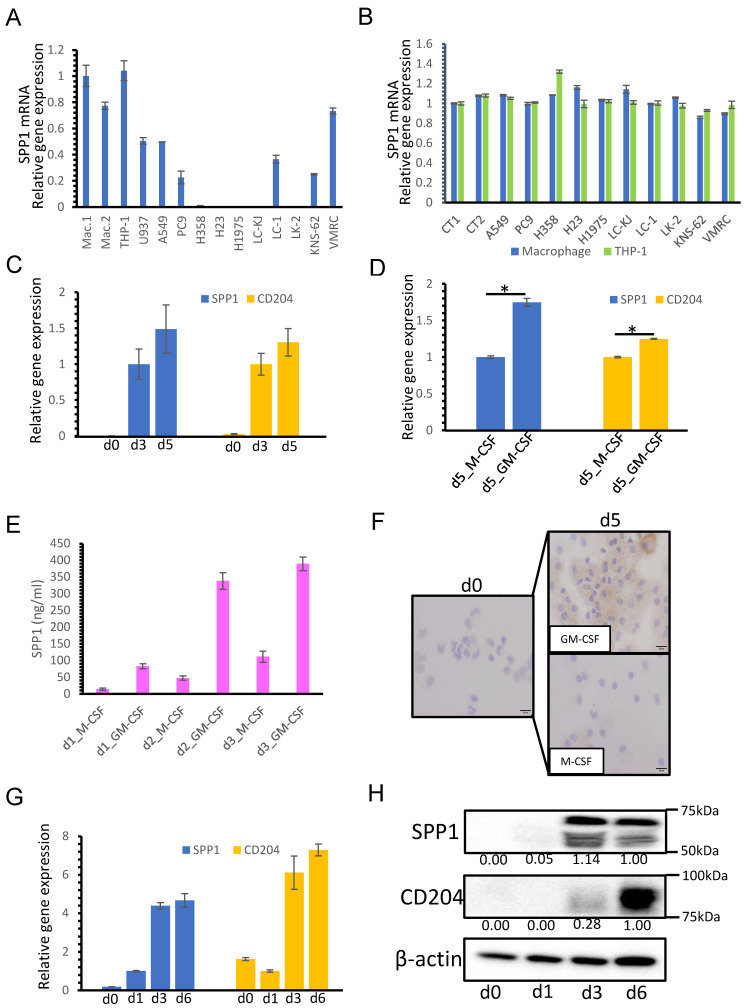
SPP1 expression in macrophages. (**A**) SPP1 expression in macrophages, THP-1 cells, U937 cells, and 10 lung cancer cell lines was examined by qRT-PCR. (**B**) Macrophages (blue) and THP-1 cells (green) were stimulated with the CM of cancer cell lines; then, the SPP1 mRNA expression level was evaluated. Control (CT) refers to macrophages or THP-1 without CM of cancer cell lines added. (**C**) Monocytes were cultured with M-CSF and GM-CSF as described in the Materials and Methods section, and SPP1 mRNA and CD204 mRNA levels were evaluated at different stages of monocyte/macrophage culture (days 0, 3, and 5). (**D**) Monocytes were cultured with M-CSF or GM-CSF for 5 days, and SPP1 and CD204 expression levels were examined. (**E**) SPP1′s concentration (ng/mL) in the CM of macrophages cultured with M-CSF or GM-CSF was examined by ELISA at days 1 to 3 (n = 3). (**F**) Immunocytostaining of SPP1 in the macrophages cultured with M-CSF or GM-CSF at days 0 and 5. (**G**) SPP1 and CD204 expression levels in THP-1 cells differentiated for 0, 1, 3, or 6 days were examined by qRT-PCR (**G**) and Western blot analysis (**H**). *: statistically significant, *p* value < 0.05.

**Figure 5 cancers-14-04374-f005:**
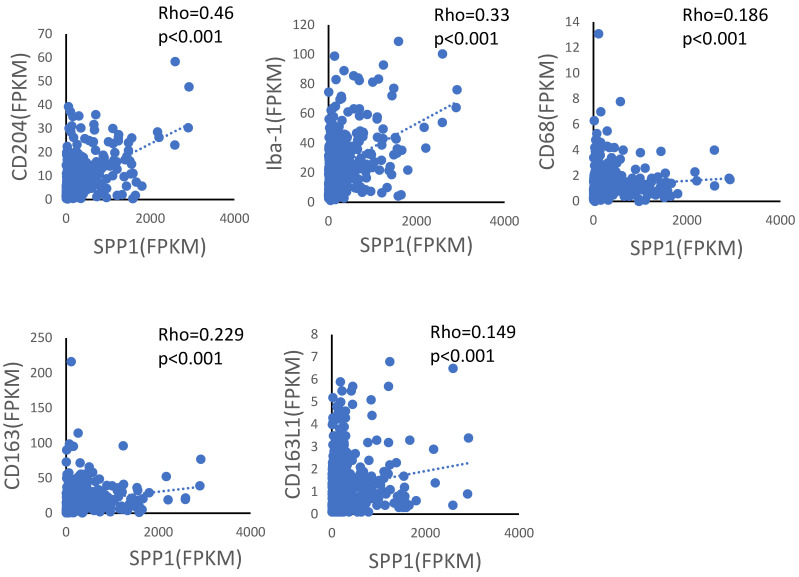
Correlation diagram between SPP1 gene expression and other macrophage markers (CD204, Iba-1, CD68, CD163, and CD163L1) in lung adenocarcinoma tissues using gene expression data from The Cancer Genome Atlas (https://www.proteinatlas.org/ (accessed on 16 June 2021)).

**Figure 6 cancers-14-04374-f006:**
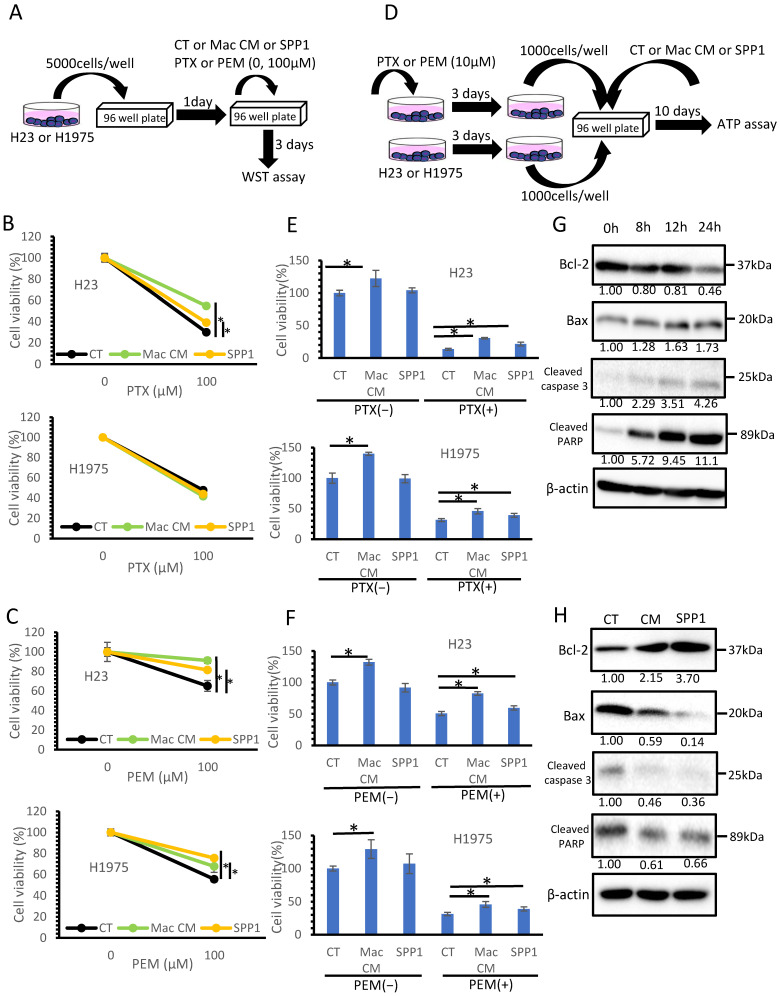
SPP1 and chemo-resistance in lung cancer cell lines. (**A**) Schema showing the 2D cell culture method for testing the chemo-sensitivity of the H23 and H1975 cell lines. Cells were cultured with or without the CM of macrophages and SPP1 and treated with PTX (**B**) or PEM (**C**). Cell viability was tested by the WST assay. (**D**) Schema showing the 3D cell culture method for testing the re-growth of the H23 and H1975 cell lines. Cells were cultured with or without the CM of macrophages and SPP1 and treated with PTX (**E**) or PEM (**F**). Cell viability was tested by an ATP assay. (**G**). Apoptosis-related molecules when PTX was added to H23 cells. (**H**) H23 cells were cultured with the CM of macrophages or SPP1 and treated with PTX for 1 day; then, the expression levels of apoptosis-related molecules were examined by Western blot analysis. CT refers to cancer cell lines without the CM of macrophages and SPP1 added. *: statistically significant, *p* value < 0.05.

**Figure 7 cancers-14-04374-f007:**
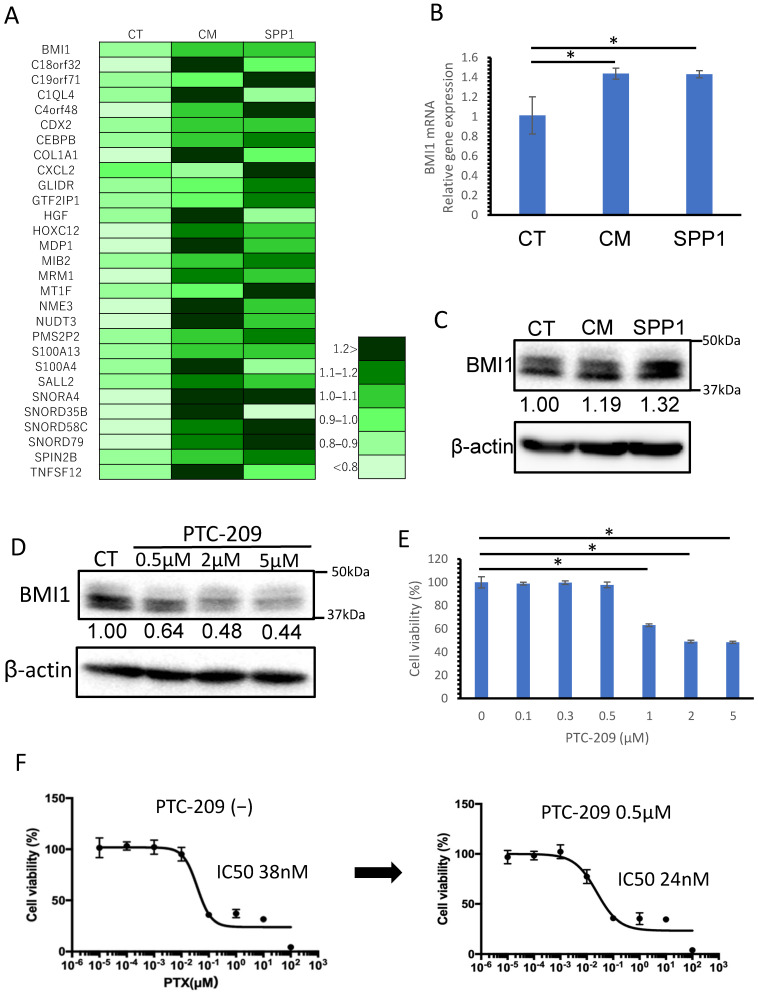
Gene expression analysis of cancer cells treated with the CM of macrophages or SPP1. (**A**) H23 cells were treated with the CM of macrophages and SPP1 for 24 h; then, the upregulated genes were detected by RNA-seq analysis (GEO: GSE212734). The upregulated genes in the cells treated with the CM of macrophages and SPP1. BMI1 expression was examined by qRT-PCR (**B**) and Western blot analysis (**C**). (**D**) H23 cells were treated with the indicated concentration of PTC-209 for 48 h, and the protein expression of BMI1 was determined by Western blot analysis. (**E**) H23 cells were treated with the indicated concentration of PTC-209 for 48 h, and cytotoxicity was examined by WST assay. (**F**) The standard curves were prepared by administering PTX to the H23 cells. The left side shows the standard curve without the administration of PTC-209, and the right side shows the standard curve under the administration of PTC-209 at a concentration of 0.5 µM. CT refers to H23 cells without the CM of macrophages and SPP1 added. *: statistically significant, *p* value < 0.05.

**Figure 8 cancers-14-04374-f008:**
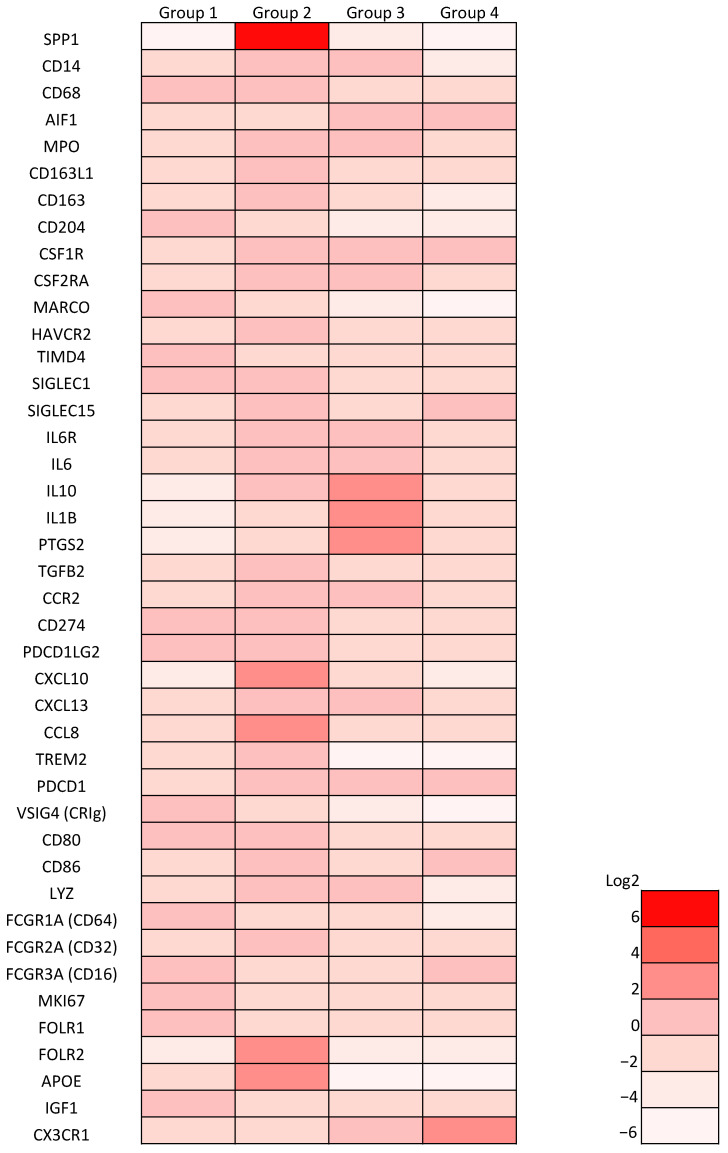
Gene expression data from the single-cell RNA-seq analysis of lung cancers published by [41].

**Table 1 cancers-14-04374-t001:** The clinical data of patients.

Variables	n (%); Ad	n (%); SCC
Age		
<65	67 (29)	20 (19)
≧65	161 (71)	83 (81)
Gender		
Male	114 (50)	93 (91)
Female	114 (50)	10 (9)
B.I.		
<600	142 (62)	14 (13)
≧600	86 (38)	89 (87)
pStage (TNM 8th)		
Stage 0	11 (5)	0 (0)
Stage IA1	41 (18)	6 (6)
Stage IA2	58 (26)	22 (21)
Stage IA3	34 (15)	17 (16)
Stage IB	32 (14)	7 (7)
Stage IIA	3 (1)	12 (12)
Stage IIB	18 (8)	19 (18)
Stage IIIA	25 (11)	14 (14)
Stage IIIB	3 (1)	6 (6)
Stage IV	3 (1)	0 (0)
Preoperative treatment		
None	223 (98)	100 (97)
Chemoradiation	5 (2)	3 (3)
Adjuvant chemotherapy		
None	167 (73)	74 (72)
UFT	28 (12)	10 (10)
Platinum based regimen	33 (15)	19 (18)
EGFR mutation		
negative	109 (48)	
positive		
L858R	54 (24)	
19del	46 (20)	
other	8 (3)	
not submitted	11 (5)	

UFT: adjuvant oral drug (generic name: tegafur-uracil).

**Table 2 cancers-14-04374-t002:** TAM-SPP1 expression and clinicopathological factors; adenocarcinoma.

		TAM-SPP1 Expression (Low: Score 0–1, High: Score 2)		
		Low	High	*p*
Age				
	<65	56	11	0.22
	≥65	121	40	
Gender				
	Male	88	26	1.00
	Female	89	25	
Smoking				
	B.I. < 600	110	32	1.00
	B.I. ≥ 600	67	19	
pStage				
	0–I	137	39	1.00
	II–VI	40	12	
EGFR mutation				
	negative	85	24	0.84
	positive	82	26	

Chi-square test was performed.

**Table 3 cancers-14-04374-t003:** TAM-SPP1 expression and clinicopathological factors; SCC.

		TAM-SPP1 Expression (Low: Score 0–1, High: Score 2)		
		Low	High	*p*
Age				
	<65	6	14	0.01
	≥65	53	30	
Gender				
	Male	54	39	0.74
	Female	5	5	
Smoking				
	B.I. < 600	6	8	0.37
	B.I. ≥ 600	53	36	
pStage				
	0–I	29	23	0.90
	II–VI	30	21	

Chi-square test (Age, Smoking, pStage) and Fisher’s exact test (Gender) was performed. Underline indicates statistically significant.

**Table 4 cancers-14-04374-t004:** Univariate and multivariate analyses for cancer-specific survival in lung adenocarcinoma.

		Univariate			Multivariate		
		*p*-Value	HR	95% CI	*p*-Value	HR	95% CI
Age	<65 vs. 65≦	0.6779	1.240	0.466–3.876			
Gender	Female vs. Male	0.0060	4.098	1.468–14.46	0.6636	1.339	0.379–5.718
Smoking	B.I. < 600 vs. 600 ≦ B.I.	0.0049	3.871	1.502–11.12	0.7926	1.171	0.373–4.167
pStage	pStage 0–I vs. pStage II–IV	<0.0001	14.59	5.222–51.56	<0.0001	10.81	3.673–39.99
EGFR	mutation vs. wild type	0.0002	8.735	2.485–55.24	0.0086	5.773	1.494–38.31
TAM-SPP1	Low score vs. High score	0.0335	3.008	1.096–7.735	0.0172	3.599	1.273–9.725

HR: hazard ratio; CI: confidence interval. Underline indicates statistical significance.

**Table 5 cancers-14-04374-t005:** Univariate and multivariate analyses for progression-free survival in lung adenocarcinoma.

		Univariate			Multivariate		
		*p*-Value	HR	95% CI	*p*-Value	HR	95% CI
Age	<65 vs. 65≦	0.5983	1.170	0.661–2.188			
Gender	Female vs. Male	0.0265	1.828	1.072–3.179	0.9458	0.978	0.519–1.862
Smoking	B.I. < 600 vs. 600 ≦ B.I.	0.0016	2.362	1.389–4.038	0.0826	1.735	0.931–3.284
pStage	pStage0–I vs. pStage II–IV	<0.0001	9.886	5.712–17.57	<0.0001	9.055	5.185–16.23
EGFR	mutation vs. wild type	0.0584	1.673	0.981–2.909			
TAM-SPP1	Low score vs. High score	0.1822	1.519	1.812–2.693			

HR: hazard ratio; CI: confidence interval. Underline indicates statistically significant.

## Data Availability

The data presented in this study are available in this article and in Appendix A.

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
