# Peer review of "SPP1 Derived from Macrophages Is Associated with a Worse Clinical Course and Chemo-Resistance in Lung Adenocarcinoma"

_cancers, 2022, doi:10.3390/cancers14184374_

Round 1

Reviewer 1 Report

The paper “ SPP1 derived from macrophages is associated with a worse clinical course and chemo-resistance in lung adenocarcinoma” by Matsubara et as is dedicated to the study of prognostic value of SPP1 in non-small cell lung cancer. Further the authors studied SPP1 expression on tumor cells and macrophages and demonstrated that specifically macrophage-derived SPP1 is prognostically significant. The authors also investigated possible mechanism of SPP1 action on tumor cells in vitro. The study is novel and interesting for a broad scientific community. 

However there are several shortcomings that hinder publication of the study in its current form. 

Major comments 

1.     The authors failed to cite a paper (https://doi.org/10.1111/1759-7714.14108) where the prognostic value of macrophage-derived SPP1 for lung cancer was demonstrated. Further the authors have to cite a paper demonstrating the usage of PU.1 as a pan-macrophage marker in IHC studies (https://doi.org/10.1155/2020/5424780). 

2.     Statistical methods used should be better described in the methods section.

3.     Survival analysis depending on the disease stage should be performed.

4.     The data of transcriptome analysis should be submitted to any public database

5.     The quality of IHC figures is poor. Better resolution is needed.

6.     Abbreviation CM is not explained anywhere. Assuming this is “conditioned medium” it is important to describe how it was generated. What was the cell density, duration of cultivation etc. 

7.     On figure 4 it is unclear how many biological replicas of each experiment were done. N has to be provided for all results presented. ELISA experiment seems to be performed only once, since no SD or SEM is indicated on the diagram.

Minor comments

1.     It is unclear from the methods section how M-CSF and GM-CSF were used in combination or as single stimuli. Results are presented for individual stimulations. 

2.     The concentration of M-CSF has to be justified, since usually 10 ng/ml of this cytokine is used. 

3.     ddCt method requires that the efficiency of both PCR reaction is the same. Was it controlled in this study?

Author Response

To Reviewer 1

We sincerely appreciate for kind reviewing and adequate suggestions. We corrected and added some sentences according to your comments.

Best regards

Eri Matsubara, Yoshihiro Komohara

Response to Reviewer 1 Comments

Major comments

  1. The authors failed to cite a paper (https://doi.org/10.1111/1759-7714.14108) where the prognostic value of macrophage-derived SPP1 for lung cancer was demonstrated. Further the authors have to cite a paper demonstrating the usage of PU.1 as a pan-macrophage marker in IHC studies (https://doi.org/10.1155/2020/5424780).

Response 1: I added the following sentences and Supplementary table 4 in discussion section.

→Li Y et al. reported that TAM-derived SPP1 upregulated PD-L1 expression in NSCLC cells, and it causes poor prognosis [https://doi.org/10.1111/1759-7714.14108]. Since we previously reported the PD-L1 expression in cancer cells and TAMs separately [https://doi.org/10.1111/cas.14128, doi: 10.1007/s00262-022-03187-4] by using same cohort of lung cancer cases. We tested if SPP1 expression of TAMs would be associated PD-L1 expression of cancer cells and TAMs in this lung cohort. However, there was no correlation between SPP1 expression of TAMs and PD-L1 expression of TAMs or cancer cells in lung adenocarcinoma (Supplementary table 4).

Supplementary table 4 can be got in the attachment.

As for the PU.1, I added the following sentence in 2.3. IHC section.

→Anti-PU.1 is used as a pan-macrophage marker and it is positive in the nucleus [https://doi.org/10.1155/2020/5424780].

  1. Statistical methods used should be better described in the methods section.

Response 2: I added the following sentences in 3.0. Statistical analysis section.

→Pearson’s chi-square test or Fisher’s exact test was used to analyze the correlation between SPP1 expression and clinical pathologic features. Cancer specific survival (CSS) was defined as the time from surgery to death for cancer. Progression-free survival (PFS) was defined as the time from surgery to occurrence of cancer progression or death. CSS and PFS were analyzed using the Kaplan-Meier methods and compared by Log-rank test and Wilcoxon test. Multivariate analyses were performed by Cox‘s proportional hazard model. The comparisons between two groups were analyzed with a student’s t-test, or a Mann-Whitney U test. Spearman correlation analysis was used for correlation analysis. For all analyses, p < 0.05 was considered to indicate statistical significance

  1. Survival analysis depending on the disease stage should be performed.

Response 3: I added the following sentence and Supplementary figure 2 in Results section (3.2).

→The CSS and PFS depending on the disease stage in adenocarcinoma are shown in Supplementary figure 2.

Supplementary figure 2 can be got in the attachment.

  1. The data of transcriptome analysis should be submitted to any public database

Response 4: We are registering the data for GEO currently. We are going to upload the data later.

  1. The quality of IHC figures is poor. Better resolution is needed.

Response 5: I apologize for my quality of IHC figures. Resolution of original figure is better.

  1. Abbreviation CM is not explained anywhere. Assuming this is “conditioned medium” it is important to describe how it was generated. What was the cell density, duration of cultivation etc.

Response 6: I added the following sentences in 2.4 Cell culture of macrophages and cancer cell lines.

→For preparation of the conditioned medium (CM) of cell lines or macrophages, cancer cells (80% confluent) or differentiated macrophages cultured new medium for 1day, and supernatant were collected.

  1. On figure 4 it is unclear how many biological replicas of each experiment were done. N has to be provided for all results presented. ELISA experiment seems to be performed only once, since no SD or SEM is indicated on the diagram.

Response 7: I added the error bars in Figure 4E, and added “n=3” in Figure 4E legends.

Figure 4E can be got in the attachment.

As for qRT-PCR, the following sentences are shown in 2.6 qRT-PCR analysis section.

“A minimum of three individual samples was used, and the expression levels were calculated from at least two technical replicates.”

Minor comments

  1. It is unclear from the methods section how M-CSF and GM-CSF were used in combination or as single stimuli. Results are presented for individual stimulations.

Response 1: In 2.4.Cell culture of macrophages and cancer cell lines section, we described “These monocytes were plated in UpCELL 6-well plates (5 × 105 cells/well; CellSeed, Tokyo, Japan), and cultured in AIM-V medium (Thermo Fisher, Waltham, MA) supplemented with 2% human serum, granulocyte-macrophage colony-stimulating factor (GM-CSF; 1 ng/mL; Wako), and macrophage-colony stimulating factor (M-CSF; 100 ng/mL; Wako) to induce macrophage differentiation.” However, in Figure 4D, 4E, 4F, we use M-CSF or GM-CSF for comparison of these.

  1. The concentration of M-CSF has to be justified, since usually 10 ng/ml of this cytokine is used.

Response 2: Our lab has used M-CSF at concentration of 100ng/ml as previously described.

  • Shinchi, Y.; Komohara, Y.; Yonemitsu, K.; Sato, K.; Ohnishi, K.; Saito, Y.; Fujiwara, Y.; Mori, T.; Shiraishi, K.; Ikeda, K.; et al. Accurate expression of PD-L1/L2 in lung adenocarcinoma cells: A retrospective study by double immunohistochemistry. Cancer Sci. 2019, 110(9), 2711-2721.
  • Shinchi, Y.; Ishizuka, S.; Komohara, Y.; Matsubara, E.; Mito, R.; Pan, C.; Yoshii, D.; Yonemitsu, K.; Fujiwara, Y.; Ikeda, K.; et al. The expression of PD-1 ligand1 on macrophages and its clinical impacts and mechanisms in lung adenocarcinoma. CancerImmunol Immunother. 2022, 1-17. doi: 1007/s00262-022-03187-4.
  • Matsubara, E.; Komohara, Y.; Shinchi, Y.; Mito, R.; Fujiwara, Y.; Ikeda, K.; Shima, T.; Shimoda, M.; Kanai, Y.; Sakagami, T.; et al. CD163-positive cancer cells are a predictor of a worse clinical course in lung adenocarcinoma. Pathol.Int. 2021, 71(10), 666-673. doi: 10.1111/pin.13144.
  1. ddCt method requires that the efficiency of both PCR reaction is the same. Was it controlled in this study?

Response 3: We used β-actin for control as described in 2.6. qRT-PCR analysis section.

Reviewer 2 Report

Dr. Matsubara and his colleagues submitted the article 'SPP1 derived from macrophages is associated with a worse 2 clinical course and chemo-resistance in lung adenocarcinoma'. 

Pros/highlights:

1. The results of the study are described in detail. The use of figures and tables is appropriate. Figure 3 and Table 5 are clearly explained.

2. The conclusions drawn are in-line with the results (no contradictions noted).

3. In the conclusion, authors are able to link the results of this cell-line/basic research study to a possible therapeutic target in NSCLC, thus, making SPP1 not only a prognostic but also a possibly predictive marker. 

Suggestions:

1. The articles cited (citation number 1 and 2) in the introduction paragraph (lines 43, 46, 47, 50) are not highly pertinent. Consider citing national statistics for this content.

2. Several acronyms have been used throughout the article without explanation (ex: UTF, PTX, PEM, CM, CT). Given the international access to the journal, consider providing details for the acronyms to help the universal readers.

3. In figure 3A, the CSS was significantly shorter in patients with higher SPP1 on TAMs despite having no statistical significant difference in PFS. Consider providing a possible explanation for this in the discussion.

4. Lines 431-435 are the same as in the conclusion (436-441). Consider removing them to avoid repetition.

Author Response

To Reviewer 2

We sincerely appreciate for kind reviewing and adequate suggestions. We corrected and added some sentences according to your comments.

Best regards

Eri Matsubara, Yoshihiro Komohara

Response to Reviewer 2 Comments

Suggestions:

  1. The articles cited (citation number 1 and 2) in the introduction paragraph (lines 43, 46, 47, 50) are not highly pertinent. Consider citing national statistics for this content.

Response 1: I revised the following sentences in the introduction paragraph.

Lung cancer is the most common cause of cancer death in the Japanese population, and its incidence increases sharply after the age of 50 years [1,2]. Although recent advancements and the widespread use of computed tomography scanning have contributed to the early diagnosis of lung cancer, about two-thirds of lung cancer patients are still diagnosed at advanced stages [3,4]. Non-small cell lung cancer (NSCLC) accounts for about 85% of all lung cancers [5]. The treatment options for NSCLC depend mainly on the stage of the cancer. Surgery to remove the cancer is an option for early stage NSCLC, and it provides the best chance to cure the disease. However, postoperative recurrence often occurs even in cases that were able to receive radical resection [1-3].

  1. Goya, T; Asamura, H.; Yoshimura, H.; Kato, H.; Shimokata, K.; Tsuchiya, R.; Sohara, Y.; Miya, T.; Miyaoka, E. Prognosis of 6644 resected non-small cell lung cancers in Japan: a Japanese lung cancer registry study. 2005, 50(2), 227-234. doi: 1016/j.lungcan.2005.05.021.
  2. Asamura, H.; Goya, T.; Koshiishi, Y. ; Sohara, Y.; Eguchi, Y.; Mori, M.; Nakanishi, Y.; Tsuchiya, R.; Shimokata, K.; Inoue, H.; et al. A Japanese Lung Cancer Registry study: prognosis of 13,010 resected lung cancers. J Thorac Oncol. 2008, 3(1), 46-52. doi: 10.1097/JTO.0b013e31815e8577.
  3. Hoffman, PC.; Mauer, AM.; Vokes, EE. Lung cancer. Lancet. 2000, 355(9202), 479-485. doi: 10.1016/S0140-6736(00)82038-3.
  4. Altorki, ; Kent, M.; Pasmantier, M. Detection of early-stage lung cancer: computed tomographic scan or chest radiograph?. J Thorac Cardiovasc Surg. 2001; 121(6), 1053-1057. doi: 10.1067/mtc.2001.112827.
  5. Mountain, CF. Revisions in the International System for Staging Lung Cancer. Chest. 1997, 111(6), 1710-1717. doi: 10.1378/chest.111.6.1710.

Several acronyms have been used throughout the article without explanation (ex: UTF, PTX, PEM, CM, CT). Given the international access to the journal, consider providing details for the acronyms to help the universal readers.

Response 2:

I added “ UFT: adjuvant oral drug (generic name: tegafur-uracil)” under the Table 1.

As for CM, I added these sentences in 2.4. Cell culture of macrophages and cancer cell lines section. “For preparation of the conditioned medium (CM) of cell lines or macrophages, cancer cells (80% confluent) or differentiated macrophages cultured new medium for 1day, and supernatant were collected.”

As for PTX and PEM, we described “Paclitaxel (PTX) and pemetrexed (PEM)” in 2.4. Cell culture of macrophages and cancer cell lines section.

As for CT, we added the following each sentence in Figure legends. In Figure 4B, “Control (CT) refers to macrophages or THP-1 without CM of cancer cell lines added.” In Figure 6, “CT refers to cancer cell lines without the CM of macrophages and SPP1 added.” In Figure 7, “CT refers to H23 cells without the CM of macrophages and SPP1 added.”

  1. In figure 3A, the CSS was significantly shorter in patients with higher SPP1 on TAMs despite having no statistical significant difference in PFS. Consider providing a possible explanation for this in the discussion.

Response 3: I added the following sentences in discussion section.

→The CSS was significantly shorter in the group with high scores for SPP1 on TAMs. As for the PFS, a similar trend was observed, but not to a statistically significant level. As a reason for that, we presumed that TAM-SPP1 high group was resistant to treatment after recurrence. However, we did not analyze that in detail, it is unclear.

  1. Lines 431-435 are the same as in the conclusion (436-441). Consider removing them to avoid repetition.

Response 4: I removed lines 431-435.

Reviewer 3 Report

I read the study by Matsubara and collaborators. The topic is current and very intriguing. SSP1 is evaluated as a prognostic factor for the course of the disease and treatment of NSCLC. However, some minor changes should be made before publication.

1-M&M 2.6 line 137

Primers shoul be insert in table. Moreover, bp, numer of access and citation of previously use should be included. If this is the first use, insert methods of design, efficiency and specificity of the primers

2-Conclusion line 440-441 Misleading claim as SPP1 is also present on cancer cells.

3- Author should increase the bibliography to give greater strength to the conclusions

Author Response

To Reviewer 3

We sincerely appreciate for kind reviewing and adequate suggestions. We corrected and added some sentences according to your comments.

Best regards

Eri Matsubara, Yoshihiro Komohara

Response to Reviewer 3 Comments

1-M&M 2.6 line 137

Primers should be insert in table. Moreover, bp, number of access and citation of previously use should be included. If this is the first use, insert methods of design, efficiency and specificity of the primers

Response 1: I added the following sentences and Supplementary table 1 in 2.6. qRT-PCR analysis section.

→The used all primers were purchased from TAKARA (Shiga, Japan), and these were pre-designed. The primers are listed in Supplementary table 1.

Supplementary table 1 can be got in the attachment.

2-Conclusion line 440-441 Misleading claim as SPP1 is also present on cancer cells.

Response 2: SPP is present both macrophages and cancer cells.

3- Author should increase the bibliography to give greater strength to the conclusions

Response 3: I added the following sentences and Supplementary table 4 in discussion section.

→Li Y et al. reported that TAM-derived SPP1 upregulated PD-L1 expression in NSCLC cells, and it causes poor prognosis [doi: 10.1111/1759-7714.14108]. Since we previously reported the PD-L1 expression in cancer cells and TAMs separately [https://doi.org/10.1111/cas.14128, doi: 10.1007/s00262-022-03187-4] by using same cohort of lung cancer cases. We tested if SPP1 expression of TAMs would be associated PD-L1 expression of cancer cells and TAMs in this lung cohort. However, there was no correlation between SPP1 expression of TAMs and PD-L1 expression of TAMs or cancer cells in lung adenocarcinoma (Supplementary table 4).

Supplementary table 4

TAM-SPP1 expression and PD-L1 expression of TAM in lung adenocarcinoma

TAM

PD-L1 low (<50%)

PD-L1 high (≥50%)

TAM-SPP1 low (Score 0, 1)

89

88

TAM-SPP1 high (Score 2)

23

28

Chi-square test was performed (p=0.62).

PD-L1 expression of TAM was evaluated based on the percentage of positive staining.

As follows: <low, 50% positive cells; high, ≥50% positive cells.

The report (doi: 10.1111/1759-7714.14108) is consistent with our observation (TAM-SPP1 predicts poor prognosis in lung adenocarcinoma), and we further investigated the correlation of TAM-SPP1 and PD-L1 expression of cancer cells or TAM.

Round 2

Reviewer 1 Report

All my comments were sufficiently addressed. No further revision is required.